# Current and Emerging Strategies to Treat Urothelial Carcinoma

**DOI:** 10.3390/cancers15194886

**Published:** 2023-10-08

**Authors:** Berkha Rani, James J. Ignatz-Hoover, Priyanka S. Rana, James J. Driscoll

**Affiliations:** 1Case Comprehensive Cancer Center, School of Medicine, Case Western Reserve University, Cleveland, OH 44106, USA; berkharani.2021@gmail.com (B.R.); james.ignatz-hoover2@uhhospitals.org (J.J.I.-H.); pxr240@case.edu (P.S.R.); 2Division of Hematology & Oncology, Department of Medicine, Case Western Reserve University, Cleveland, OH 44106, USA; 3Adult Hematologic Malignancies & Stem Cell Transplant Section, Seidman Cancer Center, University Hospitals Cleveland Medical Center, Cleveland, OH 44106, USA

**Keywords:** antibody-drug conjugate, drug resistance, immune checkpoint inhibitors, immunotherapy, metastasis, tumorigenesis, urothelial cell cancer

## Abstract

**Simple Summary:**

Urothelial cell carcinoma (UCC) is the ninth most common cancer worldwide and in the US the fourth most common cancer, with ~82,000 new cases (~62,000 men) diagnosed annually leading to ~17,000 deaths/year (~12,000 men). While early-stage cases exhibit more favorable outcomes, the emergence of drug resistance and distant metastasis reduces median overall survival (OS) to 12–15 months. The development of modern genetic and molecular assays to detect high-risk mutations has improved the detection of high-risk disease. Recently, immune therapies have been developed; these demonstrate markedly improved OS rates compared to treatment with chemotherapy alone. However, challenges persist and there remains an urgent, unmet need to develop and advance novel molecular and therapeutic strategies that prevent or overcome drug resistance, to improve patient outcome. Here, we provide an overview of the etiology, diagnostic approach and emerging therapeutic strategies for improving UCC patient quality of life and OS.

**Abstract:**

Urothelial cell carcinoma (UCC, bladder cancer, BC) remains a difficult-to-treat malignancy with a rising incidence worldwide. In the U.S., UCC is the sixth most incident neoplasm and ~90% of diagnoses are made in those >55 years of age; it is ~four times more commonly observed in men than women. The most important risk factor for developing BC is tobacco smoking, which accounts for ~50% of cases, followed by occupational exposure to aromatic amines and ionizing radiation. The standard of care for advanced UCC includes platinum-based chemotherapy and programmed cell death (PD-1) or programmed cell death ligand 1 (PD-L1) inhibitors, administered as frontline, second-line, or maintenance therapy. UCC remains generally incurable and is associated with intrinsic and acquired drug and immune resistance. UCC is lethal in the metastatic state and characterized by genomic instability, high PD-L1 expression, DNA damage-response mutations, and a high tumor mutational burden. Although immune checkpoint inhibitors (ICIs) achieve long-term durable responses in other cancers, their ability to achieve similar results with metastatic UCC (mUCC) is not as well-defined. Here, we discuss therapies to improve UCC management and how comprehensive tumor profiling can identify actionable biomarkers and eventually fulfill the promise of precision medicine for UCC patients.

## 1. Introduction

### 1.1. Overview

Urothelial cell carcinoma (UCC, formerly known as transitional cell carcinoma, TCC) is the most common neoplasm of the urinary system. UCC is the most common histologic type of BC and accounts for ~90% of all BC [1,2,3,4]. A number of histologic variants of UCC have been identified, including micropapillary, microcystic, nested, lymphoepithelioma-like, plasmacytoid, sarcomatoid, giant cell, poorly differentiated, lipid rich, clear cell and urothelial carcinoma with divergent differentiation. UCC is defined as the invasion of the cancer cells’ basement membrane or *lamina propria* or deeper by neoplastic cells of urothelial origin. Invasion is defined as ‘micro invasion’ when the depth of invasion is 2 mm or less [2]. The World Health Organization (WHO) classifies BCs based on differentiation as low grade (grades 1 and 2) or high grade (grade 3) [1]. Distinction between low-grade and high-grade urothelial disease has implications related to risk stratification, patient management and treatment outcome.

Urothelial tumors arise and evolve through divergent genetic and phenotypic pathways [2]. The advent of next-generation sequencing has allowed widespread comprehensive molecular characterization of urothelial tumors and, subsequently, the development of therapies targeting specific molecular pathways implicated in carcinogenesis, e.g., FGFR inhibition, Nectin-4, Trop-2, and HER2 targeting. As these therapies are effective in a second-line setting, they will be advanced in the treatment paradigm to localized and even non-muscle invasive disease. Some tumors progress from urothelial hyperplasia to low-grade, non-invasive superficial papillary tumors. More aggressive variants arise from flat, high-grade carcinoma in situ and progress to invasive tumors or arise de novo as invasive tumors [3,4,5]. These distinct phenotypic variants of urothelial tumors exhibit drastically different biological behavior, responses to treatment and prognoses for patients. The low-grade papillary variant is often multifocal and tends to recur, and infrequently progresses to the muscle invasive stage. In contrast, most invasive variants develop into incurable metastases despite radical cystectomy (RC) and additional modalities. It is evident that UCC variants harbor distinct genetic defects that impact growth control, metastatic potential and treatment decisions, as in Figure 1 in [6]. Low-grade, non-invasive papillary tumors are frequently characterized by activating mutations in *HRAS* and the fibroblast growth factor receptor 3 gene (*FGFR3*) [7]. High-grade invasive tumors are characterized by structural and functional defects in the *p53* and retinoblastoma protein (*Rb)* tumor-suppressor pathways [7,8].

### 1.2. Environmental and Hereditary Risk Factors for Urothelial Cancers

Risk factors for UCC are largely environmental, with smoking by far the most important. Smoking is a major risk factor for BC and accounts for ~50% of cases in both men and women. Current smokers are 4–5 times more likely to develop BC than non-smokers [9,10]. Other environmental risk factors include occupational exposure, radiation, drinking water contaminants, and chronic bladder infections [11]. These factors contribute to an increased risk of developing BC, highlighting the importance of preventative measures, e.g., smoking cessation, and minimizing exposure to harmful substances. The consequences of these risk factors and exposures leads to field changes within the urothelial tract that predispose individuals to the development of recurrent tumors, occurring in new locations in the urothelial tract, a phenomenon referred to as polychronotropism [12]. Genetics are also implicated in an individual’s susceptibility to urothelial BC. Studies have found that individuals with a family history of UCC have a higher relative risk of developing the disease. However, an increased risk cannot be entirely attributed to shared environmental exposures, indicating that genetic factors may play a role in BC development [13]. Hereditary non-polyposis colon cancer syndrome (HNPCC) is caused by mutations in DNA mismatch repair genes and can be identified through microsatellite instability (MSI) or the absence of the corresponding protein by immunostaining. HNPCC is linked to a greater risk of developing BC and upper urinary tract/urothelial cancers (UTUC).

### 1.3. Novel Approaches for Urothelial Cancers

While clinical management of early-stage UCC has seen significant improvements in the past decade, treatment of locally advanced and mUCC remains difficult [14]. Recently, a number of strategies and therapies have been developed for the treatment of advanced UCC. Noteworthy among these are immune checkpoint inhibitors (ICIs) and antibody-drug conjugates (ADCs) [15]. More traditional approaches, e.g., radiotherapy (RT), as part of trimodal therapy, is an attractive alternative treatment in patients with urothelial muscle-invasive BC (MIBC) [16,17]. Moreover, there is some evidence to suggest the effects of ICIs and RT are potentially synergistic. Photodynamic therapy (PDT) also has merit and is a promising approach. Clinical success can be limited due to the severe hypoxia which promotes ICI resistance [18]. Chemotherapy and hypoxia induce programmed death ligand-1 (PD-L1) overexpression, leading to immunosuppression within the tumor microenvironment (TME). Current approaches do not thoroughly address these defects effectively by a single drug or nano-system. To address this problem, Zhou et al. developed a biomineralization method to construct MB@Bu@MnO_2_ nanoparticles with a two-step oxygen regulation ability and PD-1/PD-L1 axis cascade-disruption capacity. Manganese dioxide albumin (MnO_2_@Alb) functions as the drug carrier, butformin (Bu) as a mitochondria-associated oxidative phosphorylation (OXPHOS) disruptor with PD-L1 depression and oxygen reversion ability, and methylene blue (MB) as a PDT drug with programmed cell death protein 1 (PD-1) inhibition capacity. Owing to the tumor-responsive capacity of MB@Bu@MnO_2_ nanoparticles, Bu and MB were selectively delivered and released in tumors. Hypoxia was reversed by Bu inhibited oxygen consumption, and MnO_2_ improved oxygen generation. Reactive oxygen species (ROS) generation was enhanced by MB@Bu@MnO_2_ nanoparticle-mediated PDT owing to reversed tumor hypoxia. The immunosuppression microenvironment was reversed by MB@Bu@MnO_2_ nanoparticles that enhanced immunogenic cell death and PD-1/PD-L1 axis cascade-disruption, which then promoted T-cell infiltration. Nanosystems could solve the defects of traditional PDT by disrupting the PD-1/PD-L1 axis and reversing the negative effects of hypoxia within the TME [19].

## 2. Divergent Mechanisms Underlying Urothelial Tumorigenesis and Treatment Resistance

### 2.1. Invasive and Non-Invasive UCC

The clinical staging of BC requires a multimodal approach that combines histopathology with molecular and imaging studies [20]. Most UCC are a form of non-muscle invasive BC (NMIBC)**.** Urothelial tumors develop along two major, largely independent but somewhat overlapping biological pathways, referred to as papillary and non-papillary or solid [21]. Current management of BC still relies on pathologic staging that does not always reflect the risk for an individual patient. Studies assessing molecular alterations in individual tumors are offering insights into the myriad of cellular pathways that are deregulated in bladder tumorigenesis and progression. Alterations in pathways involved in cell-cycle regulation, apoptosis, cell signaling, angiogenesis and tumor-cell invasion have been shown to influence disease behavior [22]. Noninvasive UCC are further subcategorized into exophytic papillary and carcinoma in situ (CIS) based upon distinct molecular alterations. Exophytic papillary tumors, or Ta tumors, which are a type of noninvasive urothelial cancer, can have varying genetic mutations and are classified as either high or low grade [23,24,25,26,27,28,29]. Low-grade Ta tumors are often associated with mutations involving receptor tyrosine kinase-*Ras* activation, specifically either *HRAS* or *FGFR3* mutations, while high-grade Ta tumors are typically linked with *p16INK4a* homozygous deletion and a lower frequency of *FGFR3* mutations [30]. In contrast, CIS and invasive tumors have alterations in *TP53* and *RB1* [31]. Muscle-invasive tumors are characterized by changes in vascular endothelial growth factors (VEGF), cadherins, and matrix metalloproteinases (MMPs) [32].

### 2.2. Cell Cycle Alterations

UCC is characterized by actionable, genetic alterations in specific molecular pathways that result in uncontrolled cellular proliferation, and that can be pharmacologically targeted [33,34,35]. As illustrated in Table 1, genetic alterations in UCC play a pivotal role in the etiology and progression of the disease. The most well-characterized pathways in UCC are those regulating the cell cycle, particularly the *p53* and *RB* mechanisms, which interact with mediators of apoptosis and gene regulation [7,8,29,36]. Thus, *TP53* is encoded on chromosome 17p13.1 and inhibits cell-cycle progression by transcriptionally activating *p21WAF1/CIP1.* Loss of *p21* expression can predict disease progression, and its maintenance can abrogate the effects of altered *TP53.* Furthermore, *TP53* mutations result in p53 inactivation and the rate of *TP53* alterations in primary tumors increases progressively from normal urothelium to non-muscle-invasive tumors, to muscle-invasive disease and metastatic nodes [37]; *TP53* is genetically altered in 49–54% of cases, while *MDM2*, which participates in an autoregulatory feedback loop with *p53*, is genetically altered in 9–11% of cases [34,38]. As a prognostic, *TP53* is not clinically established but has been shown to predict recurrence and cancer-specific mortality in muscle-invasive disease.

Rb is a cell-cycle regulatory protein that is encoded by *RB1* and that interacts with CDKs and E2F, leading to a transcription of genes required for DNA synthesis, and inactivating *RB1* mutations have been confirmed in bladder tumors [7,8,31]. *CDKN2A/B*, the genes that encode p14 and p16, inhibiting the cyclin-dependent kinases, Cdk4 and Cdk6, are altered in 5–23% of cases of MIBC [25,39]. *CCND1* (genetically altered in 10–14% of cases) and *CCND3* (genetically altered in 4–11% of cases) encode for cyclins that promote cell-cycle progression through interaction with Cdk4 and Cdk6. *CDKN1A*, a potent cyclin-dependent kinase inhibitor, is genetically altered in ~14% of cases. How CDK inhibitors, e.g., p21, p16, and p27, negatively regulate CDKs, acting as tumor suppressors is a complex process that involves multiple pathways leading to programmed cell death [38]. *Bcl-2*, an antiapoptotic member of the *Bcl-2* family of proteins, is associated with a poor prognosis in BC patients [37]. A prior study suggested that patients treated with RT for TCC of the bladder that expressed both *Bcl-2* and *TP53* had worse local control and disease-free, disease-specific OS [25]. *CDKN2A* is commonly altered in human solid tumors, but prior studies have yielded conflicting evidence regarding the association between *CDKN2A* genomic alterations and response to ICIs.

Patients harboring *CDKN2A* genomic alterations were associated with a reduced benefit from ICI therapy and were more likely to have lower PD-L1 expression in tumor-infiltrating immune cells and a less inflammatory immune-TME [39]. Palbociclib, a CDK4/6 inhibitor, was evaluated in a phase II trial of patients with metastatic, platinum-refractory UC with loss of p16 and intact Rb, and was not effective since only 17% of patients achieved PFS at 4 months [40]. Since DNA repair pathways are altered in UC, pre-clinical studies have demonstrated the effectiveness of the combination of CDK 4/6 inhibition with a PARP inhibitor [41]. Trilaciclib, another CDK 4/6 inhibitor, is being evaluated in patients with advanced/metastatic UC receiving chemotherapy followed by avelumab, which is predicted to improve outcomes by protection from myelosuppression of chemotherapy as well as enhancing the immune tumor microenvironment by inducing a transient G1 cell-cycle arrest of hematopoietic cells (NCT04887831). *CDKN2A* genomic alterations were associated with a reduced benefit from ICI therapy in UC as well as changes in the tumor-immune microenvironment [42,43].

Cell-surface receptors, which play a role in transducing external signals to the nucleus and control gene expression, are deregulated in UCC tumor cells. For example, activating *FGFR3* mutations are common in low-grade papillary Ta tumors and BC, leading to increased Ras-mitogen-activated protein kinase (*MAPK*) [42]. A large cohort of patients with metastatic urothelial cancer were treated with an anti-PD-L1 agent (atezolizumab) and major determinants of clinical outcome were identified [43,44]. Response to treatment was associated with the CD8^+^ T-effector cell phenotype, with a high neoantigen or tumor mutation burden (TMB), while lack of response was associated with a signature of transforming growth factor β (TGFβ)-signaling in fibroblasts. Other factors that impact cell signaling and gene regulation in BC include the sex hormone receptors, Janus kinase family members, and meiotic recombination 11 (*MRE11*) [45]. Reduced expression of the estrogen receptor-β (ER-β) has been linked to better progression-free survival (PFS) rates in patients with NIBC, while higher levels have been associated with more advanced tumors. Janus kinase family members, such as *STAT3,* can predict recurrence and survival in BC patients [46,47,48].

### 2.3. Influence of Angiogenesis on Invasion and Metastasis

Angiogenesis is influenced by factors secreted from tumor cells that interact with endothelial cells in the stroma. VEGFs, urokinase-type plasminogen activator (uPA), and thrombospondin 1 (TSP-1) all play a role in angiogenesis and UCC progression. Elevated levels of VEGFs and uPA have been associated with poor clinical outcomes, while TSP-1 acts as an inhibitor of angiogenesis and under expression is linked to reduced OS [47,49,50,51,52,53,54]. Reduced expression of E-cadherin has been linked to tumor recurrence and disease progression, and poor OS in BC patients [55]. Several protease families, including matrix metalloproteinases (MMPs), can also modulate the tumor’s ability to disrupt the extracellular matrix and invade neighboring tissue. Overexpression of *MMP-2* and *MMP-9* is associated with advanced-stage BC and worse OS [56,57].

### 2.4. Effect of Hypoxia on UCC Invasion and Metastasis

A major aspect in the development of various tumors, including UCC, is growth under hypoxic or/and normoxic conditions that is orchestrated by activated signaling through two major transcription factors, hypoxia-inducible factor 1-alpha (HIF-1α) and HIF-2 [58,59,60]. High immunohistochemical (IHC) expression of HIF-1α in primary UC tumors is associated with higher-grade disease, vascular endothelial growth factor-related angiogenesis, and worse prognosis with regard to disease-free and overall survival in both superficial and invasive disease. Hypoxia-induced autophagy may also propagate chemoresistance to cisplatin via the HIF-1α pathway. Resistance to ICIs presents a major obstacle in mUCC treatment and the clinical importance of hypoxia and immune status has been recently recognized [59]. The hypoxic tumor microenvironment prevents sufficient glucose oxidation, induces oxidative phosphorylation, and promotes the accumulation of lactic acid and adenosine [59,60,61]. These metabolic products lead to disruptions in T-cell ratios, T-cell inactivation, recruits immunosuppressive Tregs and decreases the invasion of antigen-presenting monocytes and dendritic cells to impair the efficacy of ICIs [62,63,64].

### 2.5. Immune Dysregulation in UCC

UCC is characterized by extensive mutational heterogeneity, which is detectable even in early-stage disease and sets the stage for the evolution of resistance [65,66]. The emergence of treatment-resistant clones is a critical barrier to cure in patients with UCC. Chemotherapy and immunotherapy both act as selective pressures that shape the evolutionary trajectory of UCC throughout the disease course. PD-L1 is a checkpoint protein that impedes immune function, allowing tumor cells to proliferate, and PD-L1 expression on tumors has been linked to advanced stage and grade, disease progression, and worse OS after cystectomy [67,68]. Tumor cells avoid immunity through immune escape, upregulation of immunosuppressive molecules, recruitment of immunosuppressive cells within the TME, and secretion of immunosuppressive signaling molecules, e.g., cytokines. Cisplatin-based chemotherapy has been the standard of care for UCC for three decades. Resistance to PD-1/PD-L1 blockade remains a significant challenge and limitation to clinical application. Defects in the antigen presentation machinery, lack of tumor antigen, T-cell dysfunction, PD-L1 negative tumor cells, non-coding RNAs that regulate tumor immunity, myeloid-derived suppressor cells (MDSCs) activity and immune checkpoints, and the gut microbiome have been implicated in patient response to anti-PD-1/PD-L1 therapy [69,70,71,72]. As a consequence, ICIs have been approved to treat advanced or metastatic BC unreponsive to Bacillus Calmette–Guerin (BCG).

## 3. Screening, Diagnostic Approach and Staging of Urothelial Carcinoma

Hematuria is the most common symptom in patients with urothelial cancer. Irritative voiding symptoms, such as dysuria, may indicate *carcinoma* in situ (*CIS*) or a bladder tumor in at-risk patients, e.g., smokers. Patients are risk-stratified, and intermediate- and high-risk patients are recommended to undergo cystoscopy to confirm the diagnosis and obtain a biopsy [73]. The staging of UCC presents a major challenge as the depth of invasion determined by cystoscopy and Transurethral Resection of Bladder Tumor (TURBT) do not uniformly correlate [74]. Although a TURBT specimen can confirm the presence of muscle-invasive (T2) disease, it cannot provide information on more advanced invasion due to the risk of bladder perforation. Where suboptimal TURBT is performed, there will be detrimental consequences on patient outcomes in regard to undergrading or understaging, increased recurrence or progression. A TURBT specimen should contain *muscularis propria*, but if absent, a repeated TURBT is often recommended. CT or non-invasive MRI can detect extravesical or nodal disease, and is more reliable if performed with a distended bladder before transurethral resection. FDG-PET/CT imaging may aid in the staging of muscle-invasive disease and detecting metastatic BC [75]. However, its usefulness in local staging is limited by the urinary excretion of FDG [75,76]. For non-muscle-invasive tumors, histologic grading is more relevant since almost all muscle-invasive neoplasms are high grade. Although there is increasing interest in utilizing molecular detection of circulating tumor cells (CTCs) in patients with UCC, the reported diagnostic effectiveness of these methods has been inconsistent. CTC detection assays for UC have low diagnostic sensitivity but high specificity for UC diagnosis. The timing of disease assessment is a crucial consideration in CTC detection since surgical interventions may result in a temporary dissemination of CTCs in the bloodstream, while chemotherapy and other systemic treatments may destroy CTCs or reduce the expression of markers, leading to a conversion of CTC-positive patients to CTC-negative [77]. Additional studies are needed to standardize techniques and determine the best marker combinations for detecting CTCs in UCC and BC patients.

## 4. Current Treatment of UCC

### 4.1. Initial Treatment

Initial treatment and prognosis of patients diagnosed with UCC depends on key factors including anatomical site, extent (stage), and histological grade. NMIBC, with the exception of *CIS*, can be treated by transurethral resection leading to excellent survival outcomes. One successful way of treating NMIBCs is using BCG vaccine intravesical immunotherapy, which decreases the risk of recurrence and progression of NMIBCs [78] Intrauterine injection of BCG causes extensive inflammation in the bladder wall, which targets tumor cells, but BCG intravesical immunotherapy may have short-term immune-stimulating effects [79]. MIBC and UTUC often require RC and/or nephroureterectomy (RNU) [80]. Systemic chemotherapy, that consists of a cisplatin-based regimen for mUCC, generally does not achieve durable responses. Therefore, the treatment outcome of patients with mUCC has been poor, with an OS rate of ~15%. Cisplatin-based neoadjuvant chemotherapy (NAC) is the standard treatment for MIBC, with benefits including downstaging and elimination of micrometastatic disease before RC [81,82]. However, patient eligibility for cisplatin-based NAC is limited by a range of factors, e.g., ECOG, creatinine clearance with hearing loss, neuropathy, and heart failure [83,84]. Tri-modality treatment therapy (TMT) involving TURBT, radiation therapy, and concurrent chemotherapy, is a commonly used alternative. In clinical trials, TMT has demonstrated 5-year OS rates comparable to NAC with RC, which yields rates of 48–65% [85]. MIBC is associated with a higher incidence of distant metastasis compared with NMIBC.

### 4.2. Bladder Preservation in UCC

TMT has demonstrated safety and efficacy in multiple studies [86,87,88]. Bladder preservation therapies, e.g., partial cystectomy TMT, are worth consideration for MIBC patients unfit for RC, ineligible for chemotherapy, or hesitant to undergo the procedure [84]. Recent studies have shown that TMT may have better cancer-specific survival and OS than RC [84,85,86,87,88,89,90]. Studies are ongoing to explore bladder preservation options for non-muscle-invasive disease (pT2 or lower) and in patients that achieve a complete response (CR) to NAC. Tumor genomic profiling technology may help to identify biomarkers that can predict response to NAC and provide treatment guidelines [91,92,93]. Clinical trials, e.g., RETAIN, are exploring active surveillance and non-surgical local therapy options for MIBC patients with specific genetic alterations and <cT1 disease on restaging TURBT [94,95].

### 4.3. Surgical

Management of UCC relies heavily on surgical intervention, particularly for early-stage disease. TURBT is the most frequently employed surgical technique for NMIBC and for patients eligible for bladder preservation therapy [91]. In more advanced cases, RC may be required [96,97]. Recent advances in surgical methods have given rise to minimally invasive approaches, e.g., robotic-assisted surgery [98,99]. Moreover, ongoing research efforts aim to develop novel surgical modalities, such as photodynamic therapy (PDT), which employs light to activate a photosensitizing agent and selectively destroy malignant cells while sparing healthy tissue [100].

### 4.4. Single Agent and Combination Chemotherapy

Patients with mUCC can benefit from a variety of chemotherapy combinations, e.g., MVAC and gemcitabine plus cisplatin [101,102]. In certain cases where combination chemotherapy is not suitable or prior treatment has failed, single-agent chemotherapy options, e.g., platinum compounds (cisplatin, carboplatin), gemcitabine, vinca alkaloids (vinblastine, vinflunine), anthracyclines (doxorubicin, epirubicin), methotrexate, taxanes (paclitaxel, docetaxel, and nanoparticle albumin-bound paclitaxel—nabpaclitaxel), and ifosfamide may be used [103,104]. However, the response to single-agent chemotherapy is brief, and there is no evidence showing improvement in survival with first-line therapy.

Worldwide, ~75% of newly diagnosed UC cases are NMIBC and the current gold standard for treatment is surgical removal by TUR. Due to the high rate (~70%) of recurrence after TUR, patients require an intensive follow-up regime that lasts many years following the initial diagnosis. This lifelong requirement for disease surveillance means UC is associated with the highest cost from diagnosis to death [38,105]. Urine cytology has high sensitivity in detecting high-grade urothelial tumors (84%), but low sensitivity in low-grade tumors (16%). The FDA has now approved six urinary assays for clinical in vitro diagnostic use: BTA-stat, BTA-TRAK, NMP22, NMP22 BladderChek, ImmunoCyt/uCyt+, and UroVysion fluorescence in situ hybridization (FISH). These tests show high sensitivity in the detection of high-grade and late-stage UC, but are unable to detect low-grade malignancies and tend to give false positive results for benign inflammatory conditions. As such, they cannot be used as stand-alone diagnostic tests for UC. Newer tests and biomarkers to stratify treatment are needed. Prior studies analyzed 118, 187 deidentified tumor samples and identified *PDL1* amplifications in 843 (0.7%), including more than 100 types of solid tumors. Most *PDL1*-amplified tumors (84.8%) had a low to intermediate TMB. *PDL1* amplification did not always correlate with high-positive PD-L1 expression by immunohistochemical analysis.

## 5. Emerging Strategies to Treat UCC

### 5.1. Targeting FGFR

Erdafitinib is a potent FGFR1-4 tyrosine kinase inhibitor. *FGFRs* are essential in regulating cell proliferation, migration, differentiation, and survival [106]. Up to 20% of patients with advanced UCC have *FGFR* alterations, and such mutations are even more frequent (37%) in patients with upper-tract urothelial carcinoma. Hence, FGFR-inhibition may be appropriate in patients with luminal I subtype disease, in which immunotherapeutic approaches may be less effective. Erdafitinib has demonstrated activity among patients with locally advanced and unresectable mUCC and with specific FGFR alterations. Erdafitinib has demonstrated activity among patients with locally advanced and unresectable mUCC and with specific FGFR alterations. In an open-label, phase 2 study, 99 patients with locally advanced or mUCC and *FGFR* alterations who had disease progression after previous chemotherapy, were enrolled. Patients were randomly assigned to receive erdafitinib in either an intermittent or a continuous regimen receiving a median of five cycles. The confirmed response rate was 40%, with a median duration of PFS of 5.5 months and OS of 13.8 months. Responses were prompt and not influenced by the number or type of prior treatments, tumor location or presence of visceral metastasis. Treatment-related adverse events (TRAEs) of ≥grade 3 were reported in 46% of patients, with 13% of patients discontinuing treatment. Common side effects included diarrhea, nausea, vomiting, fatigue, and rash, while more serious AEs were hyperphosphatemia and detachment of retinal pigment epithelium. Table 2 provides a comprehensive overview of completed and ongoing clinical trials, shedding light on the evolving landscape of treatment strategies for UCC.

### 5.2. Immune Checkpoint Inhibitors

Immunotherapy and targeted treatments are options to address situations where BCG has been unsuccessful, and for cancers that have progressed following cisplatin-based therapies (see Table 3) [107]. High TMB, DNA damage-response mutations, the presence of genomic instability and high PD-L1 expression make UCC suitable for immunotherapy. The treatment landscape for patients with advanced UCC has been significantly changed with the approval of ICIs, e.g., atezolizumab, pembrolizumab, avelumab, and durvalumab. These approvals have opened new treatment options for patients with disease progression after platinum-based chemotherapy, those who are not eligible for first-line cisplatin-based therapy and have PD-L1-positive tumors, and those who are not suitable candidates for platinum-based therapy [108,109]. In 2017, pembrolizumab, a humanized monoclonal antibody targeting PD-1, was approved for patients with advanced UCC who had progressed after cisplatin-based therapy. A phase 3 randomized, controlled trial included 542 patients with advanced UCC randomly assigned to receive pembrolizumab or the investigator’s choice of chemotherapy. The co-primary endpoints were OS and PFS. Results showed that pembrolizumab improved OS compared to chemotherapy, with a median OS of 10.3 months in the pembrolizumab group vs. 7.4 months in the chemotherapy group. The study also found fewer TRAEs of any grade in the pembrolizumab group than the chemotherapy group [110]. Interestingly, patients benefited from pembrolizumab regardless of PD-L1 expression, measured as the combined positive score (CPS), defined as the proportion of PD-L1-expressing tumor and infiltrating immune cells relative to the total number of tumor cells. PD-L1 positivity was defined as having a CPS ≥10%.

Avelumab, another ICI, received approval in 2020 as maintenance therapy for patients with locally advanced or mUCC who did not progress after first-line platinum-based chemotherapy. The results of a phase 3 trial showed that the addition of maintenance avelumab, an anti-programmed death-ligand 1 (PD-L1) monoclonal antibody, to best supportive care, prolonged OS in patients with unresectable locally advanced or mUCC who did not have disease progression with first-line chemotherapy. The study enrolled 700 patients, and OS at 1 year was significantly better in the avelumab group compared to the control group. Avelumab also significantly prolonged OS and PFS in the PD-L1-positive population. The incidence of AEs was higher in the avelumab group but generally manageable [111]. Following demonstration of efficacy, avelumab was incorporated as a treatment option for patients with advanced UCC.

After the success of ICIs in treating mUCC, they have been introduced as adjuvant therapy. In 2021, the FDA approved nivolumab for patients who have undergone complete resection and have high-risk UCC, defined as residual cancer of ≥pT2 or pN+ after receiving neoadjuvant cisplatin-based chemotherapy or ≥pT3 or pN+ without prior chemotherapy. A phase 3, multicenter, double-blind, randomized, controlled trial compared the efficacy of nivolumab vs. placebo in 709 patients with muscle-invasive UCC who had undergone radical surgery. The study showed that nivolumab significantly improved disease-free survival and survival free from recurrence outside the urothelial tract, especially in patients with a tumor PD-L1 expression level of >1% [112]. Atezolizumab, previously used in the treatment of mUCC [113], has been discontinued recently based on a phase III trial which demonstrated that, although the addition of atezlizumab to platinum-based chemotherapy resulted in improved PFS, it did not lead to any OS advantages. Consequently, the FDA revoked its regulatory approval and it is no longer used.

Despite progress in our understanding of the molecular basis of UCC, the heterogeneity of individual tumors and evolutionary pressures imposed by therapy have hampered the ability to effectively eradicate and control the disease [114,115]. Advances in our understanding of molecular mechanisms of tumorigenesis have translated into knowledge-based therapies directed against specific oncogenic signaling targets and fundamental cellular processes [116]. Recently, it was revealed that mitochondria oxidative phosphorylation (OXPHOS) depression can be used as an effective PD-L1 downregulation method. In this method, IR-LND is prepared by conjugating mitochondria-targeted heptamethine cyanine dye IR-68 with mitochondrial complexes I and II depression agent lonidamine (LND), which further self-assembles with albumin (Alb) to form IR-LND@Alb nanoparticles. The PD-L1 expression in tumors is selectively and effectively depressed by IR-LND@Alb nanoparticles. The results suggest that the antitumor efficacy of PD-L1 depression may be superior to conventional anti-PD-L1 antibodies [19,108].

### 5.3. Antibody-Drug Conjugates

Two antibody-drug conjugates (ADCs) have been approved for the treatment of BC. Enfortumab vedotin targets Nectin-4, which is overexpressed in numerous BCs and is linked to the microtubule inhibitor conjugate monomethyl auristatin E [117,118]. Enfortumab vedotin was evaluated in a global, open-label, phase 3 trial that enrolled 608 patients who had previously received platinum-containing chemotherapy and PD-1/PD-L1 inhibitors [119]. Patients were randomly assigned to receive either enfortumab vedotin or chemotherapy. Results showed that OS was longer in the enfortumab vedotin group than the chemotherapy group, with a median OS of 12.9 vs. 9.0 months, respectively. PFS was longer in the enfortumab vedotin group, with a median PFS of 5.6 months compared to 3.7 months in the chemotherapy group. The incidence of TRAEs was similar in both groups. Subsequently, it was approved by the FDA. Although skin reactions, peripheral neuropathy, and hyperglycemia are common side effects, they are mostly mild to moderate in severity. Sacituzumab govitecan is a new type of ADC that targets Trop-2 through an anti-Trop-2 humanized monoclonal antibody hRS7 IgG1κ combined with SN-38, which is the active metabolite of the topoisomerase 1 inhibitor irinotecan [120,121]. It is recognized for its distinctive toxicity profile that may lead to diarrhea, nausea, vomiting, fatigue, neutropenia, and even severe or life-threatening infusion reactions. The approval of enfortumab vedotin and sacituzumab govitecan was a significant milestone in the treatment of UCC.

### 5.4. Cellular Vaccines and Oncolytic Viruses

Vaccine therapy activates the immune system to target cancer cells using various methods, such as synthetic peptides and viral vectors. In particular, PANVAC-VF is an example of a vaccine therapy that has shown positive results in clinical trials [122], but further research is needed to optimize its dosing and effectiveness in combination with other treatments. Oncolytic viruses selectively replicate within tumor cells, resulting in their destruction and the release of additional oncolytic viruses and tumor antigens. For instance, CG0070 is an example of an oncolytic virus that specifically targets cells with RB gene mutations, which are common in UCC [123]. Despite showing promising results, the efficacy of vaccine therapy and oncolytic viruses in combination with other therapies needs further investigation.

### 5.5. CAR-T Therapy

CAR-T therapy is an emerging immunotherapy approach that involves genetically modifying a patient’s T-cells to target and attack cancer cells. In UCC, CAR-T therapy is being investigated as a potential treatment for patients with advanced or metastatic disease who have failed traditional therapies [124,125]. While still in early clinical trials, CAR-T therapy has shown improving responses in some patients with UCC. However, more research is needed to optimize the therapy and determine its long-term safety and efficacy.

### 5.6. Antiangiogenics

There is currently no established role for antiangiogenic agents in the treatment of advanced or metastatic BC. Ramucirumab and bevacizumab, both *VEGF* pathway inhibitors, have shown improved PFS but not OS in clinical trials [130,131]. Cabozantinib, a multi-kinase inhibitor including VEGF receptors, is an investigational agent for platinum-refractory mUCC as a single agent or in combination with ICIs, with promising objective response rates [132].

### 5.7. Herbal Medicines

There is accumulating, emerging information to suggest that herbal medicines could have a beneficial effect on cancer therapy and/or treatment-induced adverse side effects. Historically, natural plant-derived drugs have been considered for cancer treatment. Sanguinarine (SANG), a naturally isolated plant alkaloidal agent, possesses chemopreventive effects [133]. It has been reported that SANG impedes tumor metastasis and development by disrupting a wide range of cell signaling pathways and its molecular targets, e.g., BCL-2, MAPKs, Akt, NF-κB, ROS, and microRNAs. Additional preclinical trials may be required before these agents advance to the clinical stage.

## 6. Molecular Classification as a Prognostic to Guide Treatment Decisions

UCC are biologically, molecularly and clinically heterogeneous tumors and clinico-pathological factors fail to consistently predict treatment response and patient outcome. Molecular classifiers may help organize patients into less heterogeneous subclasses and stratify treatment decisions. Muscle-invasive BC (MIBC) is a molecularly diverse disease with heterogeneous clinical outcomes [23,134,135,136,137,138]. Table 4 lists a number of biomarkers that may have value in stratifying patient treatment decisions. Recent large-scale transcriptomic profiling studies of MIBC tumors have revealed molecularly distinct clusters. Biomarkers that are prognostic and/or predictive of therapeutic responses can be tested and validated to stratify patient treatment. Several molecular classifications have been proposed, but the molecular, genetic and histologic diversity of the disease impedes clinical application. To achieve an international consensus on MIBC molecular subtypes that reconciles the published classification schemes, the Bladder Cancer Molecular Taxonomy Group developed a classification system using transcriptomes from 1750 patients and a network-based analysis of multiple independent classification systems [24]. The primary molecular classification subdivisions were basal and luminal subtypes, and these subtypes were further subdivided by TCGA into squamous, infiltrated, luminal-papillary, luminal/genomically unstable (GU), and neuronal/small cell carcinoma (SCC) subtypes [23,24,134,135,136,137,138,139]. Each subtype is associated with unique biologic, histologic and genetic characteristics. Hence, treatment strategies may be guided by oncogenic mechanisms, immune and stromal cell infiltration, and histologic and clinical features. This TCGA analysis provided a platform to develop and guide therapies for stratified MIBC patient populations (Figure 1). Additional, prospective TCGA-based analyses are needed to address NMIBC and metastatic disease. In addition, the patient populations used to derive the current primary molecular classification were predominantly of European ancestry and it remains uncertain whether these results are applicable to other ancestral groups. The impact of ethnicity and socioeconomics may impact the biology of the disease. Prospective clinical validation of biomarkers for the six molecular subtypes could guide treatment decision-making. In addition, biomarkers within the tumor, immune cells, adjacent lymph nodes, and the TME could impact patient stratification and treatment decisions for the use of targeted therapies, ICIs, ADCs, cancer vaccines and cellular therapies.

**Table 4 cancers-15-04886-t004:** Potential Therapeutic Biomarkers to Stratify Urothelial Cancer Treatment. Compiled and adapted from references [120,126,127,128,129,140,141,142,143,144,145,146,147,148,149,150,151,152,153,154,155,156,157,158,159,160,161,162,163,164,165,166,167,168].

Chemotherapy	No. of Patients	Patient Positivity, %	Stage of Disease	Comments
DNA Damage Repair Pathways Nucleotide Excision Repair *ERCC1* Expression, *ERCC2* Mutations	100 Patients	47	Locally advanced and mUCC	mRNA obtained from patient tumor biopsies and analyzed by reverse transcriptase polymerase chain reaction (RT-PCR) was used to measure mRNA levels of several DNA repair genes. Deleterious mutations in an NER pathway helicase, *ERRC2*, are predictive of cisplatin sensitivity in bladder cancer patients. DNA repair deficiency phenotype predicts benefit from platinum-based chemotherapy.
Microsatellite Instability (MSI)	44 patients	40.6	NMIBC and MIBC	DNA from patient tumor biopsies was used to assess microsatellite sequence length using Polymerase chain reaction (PCR). MSI-H status in UC predicts deep and durable responses to CPI and is associated with inferior chemotherapy responses. CPI should be considered for first-line treatment in this subset of patients [169]
*APOBEC* mutational signature	307 patients	Up to 70	mUCC	Using patient tumor biopsies, genomic DNA was isolated and assayed using whole-exome sequencing (WES) and targeted next-generation sequencing (NGS) techniques to assess for APOBEC mutational signature. APOBEC-high are more likely to have mutations in DNA damage response genes (TP53, ATR, BRCA2) and chromatin regulatory genes (ARID1A, MLL, MLL3), potentially leading to a hypermutation phenotype and subsequent enhanced immune response against the tumor. IMvigor-130 trial- mUC patients with APOBEC mutational signature had significantly higher TMB and improved OS with atezolizumab containing regimens in the first-line cisplatin-ineligible scenario [170]
**Immunotherapy Markers**				
PD-L1—IHC	40 patients	20–72	Locally advanced and mUCC	Patient tumor biopsies were assayed for PDL1 expression using immunohistochemistry (IHC) and expressed as a combined positive score (CPS, positive tumor cells and immune cells divided by the total number of cells). PD-L1 expression and high (TMB) may predict better responses to ICIs, but patients without these biomarkers may still respond to immunotherapy. Additional caveats include a lack of standardization, tumor heterogeneity and other factors influencing the TME.
*PD-L1 IHC* *PD-L1 (CD274)*—Amplification TMB MSI	Genitourinary tumors (0.4%, 10/2420)	Protein expression- using a CPS 1 cutoff for UC, the positive prevalence was 83.6% (989/1183) Prevalence of 0.7% from 1183 patients TMB- Urothelial carcinoma (36.0%, 426/1183). MSI-H Prevalence 1.2% of 1183 patients	Locally advanced and mUCC	Retrospective pan-cancer analysis of PD-L1 immunohistochemistry and gene amplification, tumor mutation burden and microsatellite instability in 48,782 cases. analysis of all cases in which both PD-L1 IHC (using the DAKO 22C3 IHC assay with either tumor proportion score (TPS) or combined positive score (CPS); or the VENTANA SP142 assay with infiltrating immune cell score (IC)) and comprehensive genomic profiling (CGP) were tested at Foundation Medicine between January 2016 and November 2019. PD-L1 positivity was defined per the CDx indication and tumor proportion score (TPS ≥ 1) for indications without a CDx claim; and TMB positivity is defined as ≥10 mutations/Mb. A total of 48,782 cases were tested for PD-L1 IHC and CGP. Patient tumor biopsies were assayed for PD-L1 amplification using PD-L1 RNA in situ hybridization (RNAish). PD-L1 amplification was detected in only 0.7% of solid tumors. Amplification had a low correlation with PD-L1 IHC and did not correlate with TMB.
Tumor Mutational Burden (TMB)	401 patients	30.4	Locally advanced and mUCC	Genomic DNA was isolated from patient tumor biopsies and characterized for mutational burden using NGS. TMB ≥ 10 mutations per megabase was detected in 122 of 401 (30.4%) patients. Total of 191 linked to response to ICI. High TMB correlated with response in certain solid tumor types: melanoma and NSCLC. May correlate mUCC. TMB assessment is multi-factorial.
Inflammatory Gene Signatures				Checkmate-275 trial indicated a better response to Nivolumab therapy.
*ARID1A* mutation + CXCL13 expression levels	275 patients + 348 patients	50.5% 62%	Locally advanced and mUCC	DNA and RNA were isolated from patient tumor biopsies and characterized using NGS. Interrogated *CXCL13* expression and *ARID1A* mutation as a combination biomarker in predicting response to ICT in CheckMate275 and IMvigor210. The combination of the two biomarkers in baseline tumor tissues suggested improved OS compared to either single biomarker. Cumulatively, this study revealed that the combination of CXCL13 plus ARID1A may improve prediction capability for patients receiving ICT.
*TRAF2 loss*	116 patients	73%		Patient tumor biopsies were analyzed using whole-exome sequencing, RNA-seq, proteomic, and phosphoproteomic analysis to describe TRAF2 status. Proteomic analysis identified three groups reflecting distinct clinical prognoses and molecular signatures. Immune subtypes of UC tumors revealed a complex immune landscape and suggested that *TRAF2* amplification is related to the increased expression of PD-L1. Increased GARS was validated to promote the pentose phosphate pathway by inhibiting activities of PGK1 and PKM2.
*CCND1* amplification	152 patients	Primary homogeneous 15% Primary- heterogeneous 6% Metastasis- homogeneous 22% Metastasis- heterogeneous 2%	Lymph node (LN)-positive UCC pts	CCND1 and expression of CyclinD1 were evaluated by fluorescence in situ hybridization and immunohistochemistry on patient tumor biopsies obtained from node-positive urothelial bladder cancers.
**Targeted Therapies**				
*FGFR alterations*	87 patients	10–30%	Locally advanced and mUCC	DNA was isolated from patient tumor biopsies and characterized using NGS for FGFR single nucleotide variants, gene fusions or copy number abnormalities. Aberrantly activated through single-nucleotide variants, gene fusions and CNA in 5–10% of all human cancers, frequency increases to 10–30% in UC. Numerous FGFR inhibitors are currently being assessed in preclinical, Phase 1, Phase 2 and Phase 3 clinical trials. Erdafitinib and pemigatinib are currently the only approved inhibitors for use in the treatment of patients with FGFR-altered UC and Cholangiocarcinoma. UC patients with an increased frequency of FGFR3 point mutations tend to respond better to TKI therapy FGFR fusions- Clinical-grade NGS diagnostics to detect *FGFR* fusions and SNVs using tissue and ctDNA. Rapid identification of patients for targeted therapies and the real-time detection of acquired mutations that signal impending treatment resistance and cancer progression.
*NECTIN4*	169 patients	59.7%	All NMIBC and MIBC were included	Patient tumor biopsies were analyzed using IHC for Nectin-4 expression. High expression of Nectin-4 in squamous cell carcinoma and adenocarcinoma may guide treatment with novel Nectin-4-directed ADCs and provide a high-risk patient collective with a new promising therapeutic option. Nectin-4-directed therapy enfortumab vedotin, an ADC comprised of a fully human monoclonal antibody specific for nectin-4 conjugated through a cleavable linker to the microtubule inhibitor MMAE. Nectin-4 was not prognostic in histological subtypes of BC [171]
*TROP2*	>1400 patients	Up to 82%	Refractory mUCC	Patient tumor biopsies were analyzed using IHC for TROP2 expression. Sacituzumab Govitecan (SG) is an ADC targeting TROP2, approved for treatment-refractory mUC. Using gene expression data from four clinical cohorts with >1400 patient samples of muscle-invasive BC and a BC tissue microarray, we found that *TROP2* mRNA and protein are highly expressed across basal, luminal, and stroma-rich subtypes, but depleted in the neuroendocrine subtype. High levels of TROP2 in most subtypes were detected except in the neuroendocrine subtype. *TROP2* expression is higher than *NECTIN4* expression, and cells resistant to enfortumab vedotin, remain sensitive to SG.
*Erb2 (Her2, EGFR)*	128 patients	10.5%	LN-positive disease and mUCC	Patient tumor biopsies were analyzed using IHC for *ERBB2* expression. *ERBB2* overexpression and amplification were linked with high-grade and high-stage upper-tract urothelial CA (UTUCs) tumors and with tumor progression. Results suggest that ERBB2 is a biomarker for progression in UTUCs.
PI3K/AKT/mTOR/MAPK	45 patients	~42% in PI3K/AKT/mTOR 17% activating point mutations in PIK3CA 10% overexpression of AKT3 9% with mutations or deletions of TSC1/2	Refractory UC	DNA was isolated from patient tumor biopsies and mutations in mTOR genes were assayed by NGS. Limited clinical benefit with targeting this pathway in advanced UC. Phase II, single-arm, non-randomized study with everolimus in refractory UC showed minimal response with median PFS 2.6 months, median OS 8.3 months, and 2 responses seen in 45 patients.
DNA Damage Repair (DDR) Gene Abnormalities	19 patients	Alterations in DDR in up to 25%	mUCC	DNA was isolated from patient tumor biopsies and alterations in DNA damage repair genes were assayed by NGS. Single-agent olaparib showed limited antitumor activity in patients with mUC and DDR alterations. May relate to poorly characterized functional implications of particular DDR alterations and/or cross-resistance with platinum-based chemotherapy [159]
VEGF	40 patients	82%	LN-positive disease and mUCC	Patient tumor biopsies were analyzed using IHC for VEGF expression. The proportion of VEGF+ cells were defined to calculate a proportion score. The relative intensity was quantitated into an intensity score. Finally, a total immunostaining score was determined as the product of a proportion score and intensity score. Elevated VEGF correlates with worse outcome. VEGF pathway inhibition attenuates tumor proliferation and invasion. Ramucirumab, a fully humanized monoclonal antibody that binds VEGF receptor 2, has shown benefit both in randomized phase II and III trials. Results with sunitinib, pazopanib, vandetinib, or cabozantinib were not convincing.

**Figure 1 cancers-15-04886-f001:**
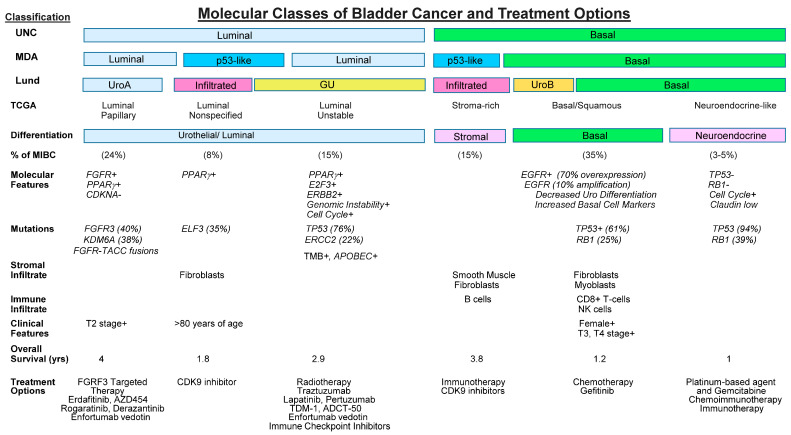
Molecular subtype classification of bladder cancer and treatment options. Figure compiled and adapted from Refs. [24,140,141,142,143,144,145,146,147,148,149,150,151,152,171,172,173,174,175,176].

Chemotherapy-resistant UCC lacks a uniformly curative therapy, and deciphering how chemotherapy directs selective pressure is a central question with clinical implications [139,177]. Therefore, to identify potential markers that correlated with treatment response, the genetic landscape of chemoradiation-treated UCC samples was analyzed [139]. Whole-exome sequencing (WES) and clonality analysis of 72 UCC samples was performed, including 16 matched sets of primary and advanced tumors prospectively collected before and after chemotherapy. Results showed that chemotherapy-treated UCC is characterized by intra-patient mutational heterogeneity, and that the majority of mutations were not shared. Mutation profiling between primary and recurrent tumors suggested that pre-existing, resistant clonal populations represented the primary mechanism of chemoradiation treatment failure. Cisplatin-based chemotherapy remains the standard of care for patients with muscle-invasive UCC but molecular determinants of response remain incompletely understood [143]. Whole-exome-sequencing was performed on pretreatment tumors and germline DNA was obtained from 50 patients with MIBC who had received neoadjuvant cisplatin-based chemotherapy followed by cystectomy (25 pT0/pTis “responders,” 25 pT2+ “nonresponders”) to identify mutations that had specifically occurred in responders. *ERCC2*, a nucleotide excision repair gene, was the only significantly mutated gene enriched in the cisplatin responders compared with nonresponders (q < 0.01). Expression of representative *ERCC2* mutants in an *ERCC2*-deficient cell line failed to rescue cisplatin and UV sensitivity compared with wild-type *ERCC2*. Somatic *ERCC2* mutations were shown to correlate with complete responses to cisplatin-based chemosensitivity in muscle-invasive UCC. Inflammation-driven phenotypic plasticity also alters the antigenic landscape of tumor cells, rewiring oncogenic signaling networks to reprogram immune cell functions [178].

A systematic meta-analysis merged gene expression datasets from the GEO repository for 18 datasets and identified 815 differentially expressed genes (DEGs) [171]. The key hub genes were screened for their differential expression in patient urine and blood plasma samples. A three-gene signature model, including *COL3A1*, *FOXM1*, and *PLK4*, was built. The predictive value regarding muscle-invasive patients’ responses to neoadjuvant chemotherapy was analyzed and a six-gene signature model, including *ANXA5*, *CD44*, *NCAM1*, *SPP1*, *CDCA8*, and *KIF14*, was developed. The study identified nine key biomarker genes—*ANXA5*, *CDT1*, *COL3A1*, *SPP1*, *VEGFA*, *CDCA8*, *HJURP*, *TOP2A*, and *COL6A1*—which were differentially expressed in urine or blood, held a prognostic or predictive value, and were immunohistochemically validated. Promising biomarkers may eventually lead to greater personalization of care for patients with BC, but remain investigational.

Five programmed death-1 (PD-1)/programmed death-ligand 1 (PD-L1) inhibitors are currently approved for treatment of locally advanced or mUCC of the bladder and the upper urinary tract [172]. Due to restrictions by the FDA and EMA, first-line treatment with Atezolizumab and Pembrolizumab in platinum-ineligible patients requires immunohistochemical PD-L1 testing. In the second-line setting, all drugs are approved without PD-L1 testing. The balance of adaptive immunity and pro-tumorigenic inflammation in individual TME is associated with PD-1/PD-L1 resistance in UCC, with the latter linked to a proinflammatory cellular state of myeloid phagocytic cells detectable in tumor and blood [173]. The quantity and spatial distribution of stromal tumor-infiltrating lymphocytes (sTIL) within the TME also predict stages of tumor inflammation, subtypes, and patient survival, and correlate with expression of immune checkpoints in an analysis of 542 patients with MIBC [174]. High sTILs indicated an inflamed subtype with an 80% 5-year DSS, and a lack of immune infiltrates identified an uninflamed subtype with a survival rate of less than 25%. A separate immune evading phenotype with upregulated immune checkpoints was associated with poor survival. Within the TIME are tertiary lymphoid structures (TLS), which can mediate antitumor activity via immune cells. High TLS amounts and close tumor distance correlated significantly with an inflamed phenotype and favorable survival. The uninflamed and evasion phenotypes showed lowest TLS numbers, farthest tumor distances, and shortest survival. High inflammation also correlated with increased neoantigen load and mutational burden.

There is a need in clinical practice to identify biomarkers that can serve as tools for risk assessment, screening and early detection of UCC, as well as for accurate diagnosis, patient prognosis, prediction of response to therapy, and cancer surveillance and monitoring response [175]. Better biomarkers are needed as tools to identify patient subsets since precision oncology is effective only in patients with specific cancer genetic mutations. Systematic, bioinformatic tools represent a valuable approach to identify and validate novel biomarkers with greater sensitivity, specificity, and positive predictive value. Publicly available genome-wide datasets, greater availability of patient pre- and post-treatment samples, analysis of tumor and immune cells, (urinary) extracellular vesicles, and the microbiome are promising sources of biomarkers. Low- or non-invasive methods of obtaining biomarkers also are appealing. Personalized treatment approaches based on individual patient characteristics and tumor biomarkers hold promise to stratify and optimize therapy and improve patient quality-of-life and OS, while reducing treatment-related toxicities. Additional candidate biomarkers to personalize BC care are under investigation, but significant obstacles must be overcome before they can be implemented in clinical practice.

## 7. Conclusions

Despite advances in diagnostic and therapeutic strategies, mUCC remains a significant challenge. The diagnostic and monitoring techniques used for urologic cancers comprise a group of invasive methodologies that still lack sensitivity and specificity [176]. Early diagnosis is crucial to improve patient outcomes and current treatments offer a range of options to manage UCC and mUCC. The treatment of mUCC has shifted dramatically following the introduction of ICIs [68,179]. Similarly, BCG is one of the most successful immunotherapies and remains the gold standard of care for patients with high-risk NMIBC, with initial response rates of ~70% [180]. Collectively, recent trials indicate that ~30% of mUCC patients demonstrate a response to ICIs. Recently approved ICIs are available as first-line therapy for cisplatin-ineligible patients or as second-line therapy for mUCC patients. Currently, selection of patients with mUCC to receive first-line immunotherapy is informed by expression of PD-L1 [181]. Patients must also be cisplatin-ineligible, but may be carboplatin-eligible if PD-L1 expression is sufficiently high. Patients with mUCC who are ineligible for any platinum-based chemotherapy may also receive immunotherapy in the first line, regardless of PD-L1 expression. High tumor mutational burden and microsatellite instability may be used to select patients for treatment with immunotherapy in cases of advanced BC, with some non-urothelial histologies in which patients have progressed. Promising innovations that may potentially advance precision medicine in UCC include multi-omic approaches, innovative trials designs, cell-free DNA, and machine learning algorithms. Biomarkers may allow for better disease classification, prognostication, and development of more efficacious noninvasive detection and surveillance strategies, as well as for the selection of therapeutic targets which can be used in BC, particularly in a neoadjuvant or adjuvant setting. Clinically relevant and useful biomarkers are also needed to identify high-risk disease, to optimize and personalize ICIs treatment for UCC. At present, alterations of *FGFR2/3* are the only genetic feature of mUCC currently used to select patients for targeted therapy [140,182,183]. In addition, erdafitinib is the only targeted therapy specifically approved for *FGFR*-altered UCC.

Recent studies have demonstrated using incorporating immunotherapies, i.e., ICIs, BCG, cytokines, adoptive T-cell immunotherapy, DCs, and macrophages, has a potential benefit for UCC patients. Nanoparticles, nano-immunotherapy, herbal medicines, and agents that overcome hypoxia-mediated immunosuppression are all potential approaches to augment PD-L1 expression on tumor cells to promote the PD-L1/PD-1 interaction. Antibiotics, fecal microbiota transplantation, and diet regulation are feasible, practical approaches to manipulate the gut microbiome and improve patient responses to immunotherapy. Prevention strategies, e.g., lifestyle modifications, smoking cessation, and chemoprevention, have the potential to reduce the incidence and improve the management and outcome of UCC. Studies that explore the utility of ICIs in the adjuvant or neoadjuvant settings, or in combination with chemotherapy, may reveal strategies to enhance patient response. The treatment paradigm for UCC is rapidly changing and novel biomarkers are emerging to select patients for appropriate therapies. Taken together, these advances offer an opportunity to improve patient survival.

## Figures and Tables

**Table 1 cancers-15-04886-t001:** Frequency of genetic alterations in select genes for MIBC tumors. Compiled and adapted from references [25,29,30,31,32,33,34,35,36,37,38,39].

Gene	Chromosomal Location	Genetic Alteration	Frequency Observed in MIBC Tumors
**Chromosome**			
	9p	Deletion	21–30%
	9q	Deletion	17%
**Oncogenes**			
*HRAS*	11p15	Activating mutation	10–15%
*FGFR3*	4p16	Activating mutation	~50% Overexpression
			15% Mutation
*PIK3CA*	3q26	Activating mutation	25%
*MDM2*	12q13	Overexpression	4% Overexpression
**Tumor suppressor genes**			
*TP53*	17p13	Deletion or mutation	70%
*RB1*	13q14	Deletion or mutation	37%
*PTEN*	10q23	Homozygous deletion or mutation	LOH 30–35%
			Mutation 17%
*CDKN2A*	9p21	Homozygous deletion or methylation or mutation	HD 20–30%
			LOH ~60%
*PTCH*	9q22	Deletion or mutation	LOH ~60%
			Mutation rare
*DBC1*	9q32–33	Deletion or methylation	LOH ~60%
*TSC1*	9q34	Deletion or mutation	LOH ~60%
			Mutation ~15%

**Table 2 cancers-15-04886-t002:** Completed and Ongoing Trials to Evaluate Novel Therapies for mUCC and MIBC. Compiled and adapted from references [19,106,107,108,109,110,111,112,113,114,115,116,117,118,119,120,121,122,123,124,125,126,127,128,129].

Trial	Patient Characteristics	Regimen	Primary & Secondary End Points	Common AEs	Results
**BLC2001** **Phase 2 study in mUCC patients**	99 patients with *FGFR* alteration, who have progressed on chemotherapy or immunotherapy	Erdafitinib 8 mg in either an intermittent or continuous regimen	Primary end point was ORR and secondary end points were PFS, OS and duration of response.	Hyperphosphatemia, Stomatitis and diarrhea.	ORR was 40%.
**KEYNOTE-045** **Phase 3 trial in mUCC**	542 patients who recurred or progressed after platinum-based chemotherapy	Pembrolizumab at a dose of 200 mg every 3 weeks or the investigator’s choice of chemotherapy with paclitaxel, docetaxel, or vinflunine	Co-primary endpoints were OS and PFS, among all patients and among patients who had PD-L1 CPS of 10% or more.	Pruritus, fatigue, and nausea	OS was 8 vs. 5.2 mos. PFS did not demonstrate a significant difference.
**JAVELIN Bladder-100—Phase 3 trial in unresectable locally advanced & mUCC**	700 patients who completed 1st line chemotherapy without progression.	Maintenance avelumab 10 mg/kg IV q2 weekly vs. best supportive care	Primary end point was OS and secondary end points included PFS and safety.	Fatigue, pruritus and urinary tract infections.	OS at 1 year was 71.3 compared to 58.4%. Median PFS was 3.7 vs. 2.0 mos.
***CheckMate-274** * **Phase 3 trial with MIBC.**	709 patients with MIBC who had undergone radical cystectomy. Neoadjuvant cisplatin-based chemotherapy before trial entry was allowed.	Adjuvant Nivolumab 240 mg IV or placebo q2 weeks for up to 1 year vs. Placebo.	Primary end point was DFS. Secondary end point was survival free from recurrence outside the urothelial tract.	Pruritus, fatigue, and diarrhea	DFS was 20.8 mos with nivolumab and 10.8 mos with placebo. Patients who were free from recurrence outside the urothelial tract at 6 mos was 77 vs. 63%.
* **EV** * ** *-* ** * **301** * **Phase 3 trial in locally advanced or mUCC**	608 patients who had previously received platinum-containing CHT and had disease progression during or after treatment with a PD-1 or PD-L1 inhibitor	Enfortumab vedotin 1.25 mg/kg on days 1, 8, 15 of a 28-day cycle or investigator-chosen CHT on day 1 of a 21-day cycle.	The primary end point was overall survival.	Alopecia, Peripheral sensory neuropathy, Pruritus.	Median OS was 12.8 vs. 8.9 mos.
**TROPHY-U- Phase II in mUCC**	113 patients who previously received platinum-containing CHT and had disease progression during or after treatment with a PD-1 or PD-L1 inhibitor	Sacituzumab govitecan 10 mg/kg on days 1 and 8 of 21-day cycles	objective response rate (ORR) secondary end points were PFS, OS, duration of response, and safety	neutropenia (3%) leukopenia (18%), anemia	ORR of 27%
**DANUBE-** **Phase 3 trial**	1032 patients that had received Durvalumab (346), Durvalumab+ Tremelimumab(342), or chemotherapy(344) in patients with untreated, unresectable or locally advance mUCC	Durvalumab (1500 mg) IV q4 weeks; Durvalumab (1500 mg)+ Tremelimumab (75 mg) IV q4 weeks for up to 4 doses, followed by durvalumab maintenance (1500 mg) q4 weeks; or SOC chemotherapy (gemcitabine + cisplatin/carboplatin) IV for up to 6 cycles.	Co-primary endpoints were OS compared b/w durvalumab and CT in pts whose tumor cells and/or tumor-infiltrating immune cells express high levels of PD-L1 (≥25%) and between durvalumab + tremelimumab and CT regardless of PD-L1 expression	Increased lipase in the Durvalumab Group and neutropenia in the chemotherapy group.	Did not meet either of the co-primary endpoints.
**Checkmate 901**	608 patients were randomized to either Nivolumab + Ipilumimab with chemotherapy or chemotherapy alone.	Nivolumab 360 mg combined with CHT every 3 weeks or CHT alone.	Dual endpoints were PFS and OS	Pruritis, fatigue, diarrhea, pneumonitis	Met its dual primary end points
**KEYNOTE-361** **Phase 3 trial**	1010 patients were randomly assigned to receive Pembrolizumab with CHT or Pembrolizumab alone or CHT alone	Pembrolizumab 200 mg q3 weeks for a maximum of 35 cycles + IV CHT on D 1 and 8 vs. CHT on day 1 of every 3-week cycle for a maximum of 6 cycles	Dual primary endpoints of OS and PFS. Secondary endpoints included duration of response, disease control rate, overall response rate and safety	Fatigue, musculoskeletal pain, decreased appetite, constipation, rash, and diarrhea	Did not meet either of the endpoints.
**DISCUS** **(ONGOING)**	224 eligible and evaluable patients (112 in each arm) to receive 3 vs. 6 cycles of platinum-based CHT + Avelumab in 1st line of mUCC	Gemcitabine on D1 and D8 with Carboplatin/Cisplatin on D1 and Avelumab every 2weekly.	QoL as measured by the change from baseline in EORTC QLQ-C30 questionnaire GHS/QoL scale scores from baseline to the completion of 6 cycles of treatment		
**Main-CAV Alliance A032001** **(ONGOING)**	Maintenance Cabozantinib+Avelumab vs. Avelumab in 1st line mUCC with clinical benefit after platinum-based CHT	Avelumab 800 mg IV q2 wk or combination of Avelumab and CABO 40 mg orally daily for up to 2 yrs	OS		

**Table 3 cancers-15-04886-t003:** Current and Emerging Therapeutic Options for the Treatment of mUCC. Compiled and adapted from references [19,106,107,108,109,110,111,112,113,114,115,116,117,118,119,120,121,122,123,124,125,130,131].

Type of Treatment	Examples	Mechanism of Action	Indications	Adverse Events
Chemotherapy	Gemcitabine, Cisplatin, Methotrexate	Kills rapidly dividing cancer cells	Advanced/metastatic stages. Contraindications include ECOG, hearing disorder, heart failure, peripheral neuropathy and Creatinine Cl <60.	Nausea and vomiting. Loss of appetite, Hair loss, Mouth sores, Diarrhea, Constipation
Immunotherapy (PD-1/PD-L1 inhibitors)	Pembrolizumab, Atezolizumab	Blocks PD-1/PD-L1 interaction, boosting immune response	Advanced/metastatic stages. Unresponsive to other treatments. Indicated as frontline for platinum-ineligible patients.	Fatigue, Nausea, Loss of appetite, Fever, Urinary tract infections (UTIs)
FGFR3 Inhibitors	Erdafitinib	Inhibits FGFR3, a gene mutation common in UCC	Advanced/metastatic with FGFR3 mutation. Progressed on prior treatment	Hyperphosphatemia, Stomatitis and diarrhea.
Antibody-Drug Conjugates	Enfortumab vedotin	Targeted delivery of toxic agents to tumor cells that express Nectin-4	Advanced/metastatic stages. Progressed on prior cisplatin-based therapy or immunotherapy.	Alopecia, Peripheral sensory neuropathy, pruritus.
CAR-T Cell Therapy	Currently not approved for mUCC	Autologous patient T-cells engineered to express a chimeric antigen receptor (CAR) directed against a cancer cell target. Potential targets include EGFR, MUC1, PD-1, HER2 and PSMA.	Under clinical investigation	Immune effector cell-associated neurotoxicity syndrome (ICANS), Cytokine Release Syndrome (CRS)

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
