# Peer review of "Current and Emerging Strategies to Treat Urothelial Carcinoma"

_cancers, 2023, doi:10.3390/cancers15194886_

Round 1

Reviewer 1 Report

Journal of Cancers

Review Article;

The article entitled “Emerging Novel Therapies to Treat Urothelial Carcinoma”. The author studies that Urothelial cell carcinoma remains a difficult-to-treat malignancy with rising incidence worldwide. The standard of care for advanced Urothelial cell carcinoma includes platinum-based chemotherapy and programmed cell death or programmed cell death ligand 1 inhibitors, administered as frontline, second-line, or maintenance therapy. Urothelial cell carcinoma is highly aggressive and remains generally incurable since these cancers are associated with intrinsic and acquired drug resistance. As Urothelial cell carcinoma is highly lethal in the metastatic state and characterized by genomic instability, high programmed cell death ligand 1 expression, DNA damage-response mutations, and high tumor mutational burden. The study discusses novel therapies to improve the management of metastatic Urothelial cell carcinoma.

I carefully read the manuscript and found it is a wonderful effort by the author but there are some minor revision needs and fulfill the mistake which the author could have done during writing.

Comments for Authors

Ø  The author needs to revise the title of the article.

Ø  Write the keyword in alphabetical order.

Ø  The author revise and includes the latest references in the introduction section.

Ø  Why did the author focus the PD-1 and PD-L1 during the study? What parameters make it important for therapy, as there are several important targets to focus the attention of Therapies to Treat Urothelial Carcinoma?

Ø  Line 78, what is ex[posures? The author needs to remove such mistakes.

Ø  The need to include the effect of hypoxia Urothelial Carcinoma.

Ø  The author needs to include the table for the treatment option agent for Urothelial Carcinoma.

Cite the following references;

v  DOI: 10.2174/1871520622666220831124321

Reviewer 2 Report

      In this research, the authors reviewed the status of Emerging Novel Therapies to Treat Urothelial Carcinoma. In my opinion, the current version of this manuscript fits the scope of Cancers and could be accepted after major revision.

My specific comments are in detail listed below:

1.     Some minor mistakes exist in the references. Besides, some references are out of date (before 2010). The authors should correct it.

2.     The authors should better reviewed the acquired immune resistance faced by in treating urothelial carcinoma, as well as emerging novel therapies for treating it, especially the possible role of PD-L1 in this. Some references should be added to this part including 10.1002/adma.202206121.

3.     In my opinion, if possible, a figure that could summaries the scope and content of this review could be added.

4.     In the introduction part, the merits of currently used tumor therapies, like PDT, RT should be better introduced in treating urothelial carcinoma. Some references should be added to this part including 10.1016/j.jconrel.2022.11.004.

5.     The clinical transformation barriers of the mentioned novel therapies for treating urothelial carcinoma should be better out-looked.

Reviewer 3 Report

The paper presented, entitled "Emerging Novel Therapies to Treat Urothelial Carcinoma," summarizes new therapeutic strategies, some of which are still experimental. It is a well-written review review.

In addition to the current standards of therapy, molecular aspects of urothelial carcinoma are addressed, which open up the new therapeutic options. However, the link from molecular aberrations to therapeutic options is partially missing. On the one hand, it was elaborated that erdafitinib is specifically active in FGFR3 alterations. On the other hand, it would be good to address which therapeutic options arise for the respective molecular subtypes of urothelial carcinoma according to the consensus classification of Kamoun et al. Are there now therapeutic strategies for CDKN2A alterations?

It would be good if issues of tolerability and side effects were addressed in the section on immune checkpoint inhibitor therapy.

Reviewer 4 Report

In my opinion, the manuscript sent for review is a paper that presents information in a very selective, not very in-depth way and it does not present any new knowledge or views . Moreover, it presents imprecise or even misleading information, such as for example in the abstract line 28: "UCC is highly aggressive", which is not consistent with observations because only about 15-20% of non-invasive cancers go into invasive stages and are associated with a short lifespan after diagnosis. The paper summarizes some of the available research, but does not present the latest findings. Regarding biomarkers, it refers to the findings of the USPSTF from 2011 and completely ignores all attempts to find markers and develop tests to help in diagnosis and treatment in both the coding and non-coding parts of DNA. In conclusion, I consider this review to be not helpful and omit many important studies available on the market. Minor comments: should be noted that the authors also incorrectly use the notation of gene names in the tables.

Round 2

Reviewer 2 Report

The current version of this manuscript could be accepted.

Author Response

We thank the Reviewer for his previous comments and agree that the manuscript can now be accepted.

Reviewer 4 Report

The authors revised the manuscript and improved its quality but still presents data without deep understanding of the subject and recent findings. Only two unsophisticated tables accompany the text and no graphical representations to facilitate quick recognition of the collected content.

Author Response

We thank the reviewer for his comments which have significantly improved the quality of the manuscript. We have included Tables 3 and 4 to address the current and emerging therapeutic options for the treatment of mUCC and potential therapeutic biomarkers to stratify UC treatment, respectively.  This includes relevant recently completed and ongoing trials. In addition, we have included Fig. 1 which illustrates the molecular classifications of BC and treatment options. 

Round 3

Reviewer 4 Report

The work is definitely more complete, but table 4 requires more details regarding the number of patients included in the referral, stage of the disease, and techniques. As it stands, this table is not ready for publication.

Author Response

Please see the attachment with the modified Table.